# Selection of Non-*Saccharomyces* Wine Yeasts for the Production of Leavened Doughs

**DOI:** 10.3390/microorganisms10091849

**Published:** 2022-09-15

**Authors:** Teresa Zotta, Tiziana Di Renzo, Alida Sorrentino, Anna Reale, Floriana Boscaino

**Affiliations:** 1Scuola di Scienze Agrarie, Alimentari, Forestali ed Ambientali (SAFE), Università degli Studi della Basilicata, 85100 Potenza, Italy; 2Institute of Food Sciences, National Research Council (CNR-ISA), Via Roma 64, 83100 Avellino, Italy

**Keywords:** bakery products, non-conventional yeasts, *Hanseniaspora* spp., leavening ability, volatile organic compounds (VOC)

## Abstract

Background: Non-conventional yeasts (NCY) (i.e., non-*Saccharomyces*) may be used as alternative starters to promote biodiversity and quality of fermented foods and beverages (e.g., wine, beer, bakery products). Methods: A total of 32 wine-associated yeasts (Campania region, Italy) were genetically identified and screened for decarboxylase activity and leavening ability. The best selected strains were used to study the leavening kinetics in model doughs (MDs). A commercial strain of *Saccharomyces cerevisiae* was used as the control. The volatile organic profiles of the inoculated MDs were analyzed by solid phase microextraction/gas chromatography-mass spectrometry (SPME/GC-MS). Results: Most of strains belonged to the NCY species *Hanseniaspora uvarum*, *Metschnikowia pulcherrima*, *Pichia kudriavzevii*, *Torulaspora delbruekii*, and *Zygotorulaspora florentina*, while a few strains were *S. cerevisiae*. Most strains of *H. uvarum* lacked decarboxylase activity and showed a high leaving activity after 24 h of incubation that was comparable to the *S. cerevisiae* strains. The selected *H. uvarum* strains generated a different flavor profile of the doughs compared to the *S. cerevisiae* strains. In particular, NCY reduced the fraction of aldehydes that were potentially involved in oxidative phenomena. Conclusions: The use of NCY could be advantageous in the bakery industry, as they can provide greater diversity than *S. cerevisiae*-based products, and may be useful in reducing and avoiding yeast intolerance.

## 1. Introduction

Yeasts play an important role in different fermentation processes. In particular, species belonging to the *Saccharomyces* genus are involved in the production of several fermented foods and beverages, ranging from alcoholic beverages (wine, beer, kefir, cider), non-alcoholic beverages (fermented tea), fermented meat and fish products, and leavened cereal-based products [1,2,3,4,5].

*S. cerevisiae* has been widely used as baker’s yeast for centuries, either alone or in combination with other yeasts and lactic acid bacteria, and most of the used strains are employed because of their fermentative behavior and technological properties that allow for uniform and standard quality products [6,7]. During fermentation, *S. cerevisiae* leads the production of CO_2_ (which results in an increase in dough volume) and the synthesis of primary and secondary metabolites, which affect the structure and sensory features of bakery products. The production of secondary metabolites (such as esters, aldehydes, ketones, organic acids, amino acids) is important because it ensures the diversity and uniqueness of the products. Therefore, strain selection is a crucial step in the bakery industry. Unconventional yeasts (i.e., non-*Saccharomyces*) have received considerable attention in the wine and brewing industry compared to the bakery leavened industry [8,9].

Some authors [10,11], however, have suggested that some species can be successfully used as alternative cultures to address the new and more demanding challenges of the baking industry. Indeed, NCY possess different metabolic features such as ability to produce aromatic compounds, antimicrobial activity, stress resistance that could favor biodiversity, and quality of foods. On the other hand, strains belonging to species other than *S. cerevisiae* have been widely found during sourdough fermentation [12]. Yeast species contribute significantly to the flavor of bakery products; conventional *S. cerevisiae* produces a limited flavor profile, whereas some NCY have demonstrated a high diversity in the production of metabolites, resulting in a more appreciable sensory quality of bread [13]. The leavening ability of NC-strains has been reported for some yeasts belonging to the species *Issatchenkia orientalis*, *Pichia membranaefaciens*, *P. kudriavzevii*, *Wickerhamomyces anomalus*, and *Torulaspora delbrueckii*. The ability to assimilate and metabolize different simple and complex sugars is widespread among NCY, suggesting that these species could provide an efficient fermentation process [13]. NCY, moreover, showed higher stress tolerance than *S. cerevisiae* strains [13]. Indeed, the fermentation performance of conventional baker’s yeast is often compromised by stress factors that occur during dough formulation, including osmotic pressure due to high sugar and salt content. The NC-species *Torulaspora delbrueckii* has also been studied for its high tolerance to freezing and thawing [14,15], two important features for the production of frozen leavened doughs. Moreover, the potential correlation between *S. cerevisiae* and some adverse reactions (i.e., inflammations, intolerance) to the ingestion of baker’s yeast in some people should not be overlooked [16]. It is known, in fact, that some people suffer intolerances due to the consumption of products that are obtained by *S. cerevisiae*-driven fermentation and the use of alternative cultures may promote diversification and safety aspects of products.

However, although the potential of some NCY during dough fermentation was already addressed by different authors [10,11], the number of tested strains was low, limiting the understanding of the functional diversity of NCY. Therefore, the study and the characterization of NCY biodiversity is necessary to select strains with useful and unique metabolic properties that are potentially interesting for the applications in the bakery industry.

Therefore, in this study we evaluated the potential of several NC-yeasts that were isolated from wine-associated sources, in order to identify strains with good leavening and aromatic abilities to be used as alternative cultures for the production of leavened doughs.

## 2. Materials and Methods

### 2.1. Strains and Culture Conditions

A total of 32 yeast strains (Table 1) that were previously isolated from grapes and must from different varieties in the Irpinia area (Campania region, Italy) were used in this study. The strains were maintained as frozen stocks (in 50% *v*/*v* glycerol) in the Yeast Culture Collection of the Institute of Food Sciences—National Research Council (ISA-CNR; Avellino, Italy) and routinely propagated in YPD broth (20 g/L bacteriological peptone, 20 g/L dextrose, 10 g/L yeast extract), for 24 h at 28 °C.

### 2.2. Biotyping and Identification of Yeast Strains

The yeast strains, previously classified at the genus level by phenotypic tests, were biotyped by RAPD-PCR (randomly amplified polymorphic DNA—polymerase chain reaction) using primers M13 [17], P80 [18], and OPA9 [19]. A representative strain from each cluster that was obtained by RAPD-PCR analysis was genetically identified by sequencing the 26S rRNA region. Total genomic DNA was extracted according to Querol et al. (1992) [20] and the quantity and purity was assessed by reading the optical density at 260 nm and 280 nm [21].

RAPD-PCR profile

The amplification mixture (25 µL) contained 10 mmol/L Tris-HCl pH 8.3, 50 mmol/L KCl, 0.2 mM dNTPs, 1.5 mM MgCl_2_, 1 µM primer, 1.25 U Taq-DNA polymerase (Biotechrabbit, Germany), and 80 ng DNA template. PCRs were performed in a Mastercycler nexus (Eppendorf, Hamburg, Germany) using the following amplification conditions: (a) for M13 (5′-GAGGGTGGCGGTTCT-3′) 35 cycles of 94 °C for 1 min, 45 °C for 20 s, ramp to 72 °C at 0·5 °C/s, 72 °C for 2 min; (b) for P80 (5′-CGCGTGCCCA-3′) an initial step of 94 °C for 2 min, followed by 40 cycles of 94 °C for 1 min, 40 °C for 1 min, 72 °C for 1 min and 30 s, and a final step at 72 °C for 10 min; (c) for OPA9 (5′-GGGTAACGCC-3′) an initial step of 94 °C for 2 min, followed by 40 cycles of 94 °C for 1 min, 36 °C for 1 min, 72 °C for 1 min and 30 s, and a final step at 72 °C for 10 min.

The amplification products were separated (90 min at 100 V) on 1.5% (*w*/*v*) agarose gel (Sigma-Aldrich, Steinheim, Germany), stained with 50 μg/mL GelRed^TM^ (10,000× in water; Botium Inc., Fremont, CA, USA), and visualized using a GelDoc XR system (Bio-Rad Laboratories, Hercules, CA, USA). The RAPD-PCR profiles were analyzed with Gel Compare II v.6.6 software (Applied Maths, Sint-Martens-Latem, Belgium), and a hierarchical cluster analysis (unweighted pair group method using average linkage, UPGMA) was carried out on the Pearson’s similarity matrix of the band profiles.

Species-level identification

Representative strains of each RAPD-PCR cluster were selected and identified by sequencing the 26S rRNA region. DNA from the selected strains was amplified using primers NL1 (5′-GCCATATCAATA AGCGGAGGAAAAG-3′) and NL4 (5′-GGTCCGTGTTTCAAGACGG-3′) according to the protocol of Kurtzman and Robnett (1998) [22]. After purification (QIAquick PCR purification kit, QIAGEN GmbH, Hilden), the amplicons were sequenced (Eurofins Genomics, Ebersberg, Germany) and sequence homology and identification were carried out using a BLAST comparison on the GenBank database (https://blast.ncbi.nlm.nih.gov/Blast.cgi; accessed on 1 May 2019).

### 2.3. Screening of the Decarboxylase Activity of Amino Acids

The decarboxylation of amino acids was evaluated according to Tristezza et al. (2013) [23]. Yeast cultures (28 °C, 24 h, from YPD broth) were standardized at 10^6^ cfu/mL and inoculated (10 µL/spot) onto YPD agar plates containing 1% (*w*/*v*) arginine, or histidine, leucine, lysine, phenylalanine, tryptophan, or tyrosine, and 0.006% (*w*/*v*) bromocresol purple (BCP) as a pH indicator. Unsupplemented YPD (without amino acids) was used as a control. The YPD plates were incubated for 7 days at 28 °C, and the appearance of purple halos (positive results) around the colonies was checked at 24 h intervals. A cell suspension (10^6^ cfu/mL) of the fresh commercial yeast *Saccharomyces cerevisiae* Lievital, (Lesaffre Italia SpA) was used for comparison.

### 2.4. Screening for the Leavening Capability in Model Wheat Doughs

Cultures (28 °C, 24 h, from YPD broth) of the 32 strains were standardized to 10^7^ cfu/mL and used to inoculate model wheat doughs (10 mL/100 g dough, 10^6^ cfu/g final population). A commercial wheat flour for breadmaking was used in all tests, with a protein content of 14% (*w*/*w* d.m.), 82% carbohydrate (*w*/*w* d.m.), and 4% fat (*w*/*w* d.m.) (F.lli De Cecco factory—Fara S. Martino, Chieti, Italia). The doughs were prepared by mixing (5 min) sterile water, flour, NaCl (2% *w*/*w*), and cell suspension to obtain a dough yield (DY) of 170. A dough that was inoculated with the commercial yeast *S. cerevisiae* Lievital (10^6^ cfu/g of dough) was used as a control. After mixing, the model doughs were portioned into Petri dishes (15 g/each) and incubated at 28 °C for 24 h. After 0 h, 6 h, and 24 h incubation, the leavening capability was evaluated by scanning the Petri dishes and measuring the difference in the dough areas [24] with the ImageJ program (https://imagej.nih.gov/ij/; accessed on 1 March 2020). The pH values were also evaluated (pH-meter BASIC 20; Crison Instruments, Barcelona, Spain). There were two biological replicates that were prepared for each strain and for the control dough.

### 2.5. Leavening Kinetics in Model Wheat Doughs

A total of six strains were selected and used to evaluate the leavening kinetics in model wheat doughs. Standardized cell suspensions and doughs were prepared as described in Section 2.4, and samples were incubated at 28 °C for 24 h. After 0, 2, 4, 6, 8, and 24 h of incubation, the leavening capability and pH values were measured as described in Section 2.4. and the final yeast populations (at 24 h) were evaluated by plate count on YPD agar (28 °C for 48 h). The commercial yeast *S. cerevisiae* Lievital was used as a control. There were three biological replicates that were prepared for each sample.

### 2.6. Measurement of Volatile Organic Compounds in Model Wheat Doughs

The most interesting strains, selected on the basis of the performance that was described in the previous sections (Section 2.3, Section 2.4, Section 2.5) were evaluated for the ability to affect the aroma profile of the doughs after 24 h fermentation. The doughs were prepared as described in Section 2.4 and the commercial yeast *S. cerevisiae* Lievital was used as a control. The volatile organic compounds (VOCs) of the inoculated doughs were measured after 24 h of incubation according to Reale et al. (2016) [25].

VOCs were extracted from doughs by using the Solid Phase MicroExtraction (SPME) and analyzed with gas chromatography coupled to mass spectrometry (SPME/GC-MS). Briefly, 2 g of samples were placed into a 20 mL headspace vial, and 5 μL of 4-methyl-2 pentanol (internal standard, 100 mg/L standard solution) was added. The vial was placed in a thermostatic block (40 °C) on a stirrer, the fiber was inserted and maintained in the sample headspace for 30 min, then removed and immediately inserted into the GC/MS injector for the desorption of compounds. The extraction was performed automatically by the multipurpose sampler of the GC/MS system. A silica fiber, coated with 75 μm of Carboxen/Polydimethylsiloxane (CAR/PDMS (Supelco, Bellefonte, PA, USA) was used for analysis. The SPME-GC/MS analysis was performed using an Agilent GC 7890A/MSD 5975 system with a Gerstel MPS2 autosampler (Agilent Technologies, Santa Clara, USA); the operating conditions were as follows: HP-Innowax capillary column (Agilent Technologies, 30 m × 0.25 mm ID, film thickness 0.25 μm), gas carrier was helium (flow 1.5 mL/min), and SPME injections were splitless (straight glass line, 0.75 mm ID) at 240 °C for 20 min, during which time thermal desorption of the analytes from the fiber occurred. The oven parameters were as follows: initial temperature of 40 °C held for 3 min, followed by an increase to 240 °C at a rate of 5 °C/min, and then held for 10 min. The injector temperature was 240 °C. The mass spectrometer operated in scan mode over a mass range from 33 to 300 amu (2 s/scan) at an ionization potential of 70 eV. VOCs identification was achieved by comparing mass spectra with the Wiley library (Wiley7, NIST 05). The data were expressed as the relative peak area with respect to the internal standard. Blank experiments were conducted in two different modalities: blank of the fiber and blank of the empty vial. These types of control were carried out after every 10 analyses. There were two technical replicates that were carried out for each sample.

### 2.7. Statistical Analysis

All graphs were performed using Systat 13.0 for Windows (Systat Software Inc., San Jose, CA, USA). One-way ANOVA and Tukey’s test were used to determine significant differences between VOCs. Principal component analysis (PCA) was performed to find correlations between VOCs and doughs that were leavened with different yeasts.

## 3. Results and Discussion

The results of RAPD-PCR analysis are shown in Figure 1. The primers that were used generated between 2 and 8 amplicons of amplified DNA fragments and an amplified fragment size between 200 and 2000 bp. Considering the genetic relationships between the strains with more than 75% similarity of the RAPD profiles, ten clusters were distinguished (named A, B, C, D, E, F, G, H, I, L) and for each cluster a representative strain was sequenced from the 26S rRNA region for identification purposes (marked with an asterisk in Figure 1). The results of the sequencing analysis are reported in Appendix A.

The clustering of the RAPD-PCR profiles and sequencing of the 26S rRNA region allowed all 32 yeast strains to be identified.

Most of the strains (22) belonged to the *Hanseniaspora uvarum* species, while some isolates matched to the *S. cerevisiae* species (6). One strain of each of the following species was also identified: *Metschnikowia pulcherrima* (1), *Pichia kudriavzevii* (1), *Torulaspora delbruekii* (1), and *Zygotorulaspora florentina* (1).

**Figure 1 microorganisms-10-01849-f001:**
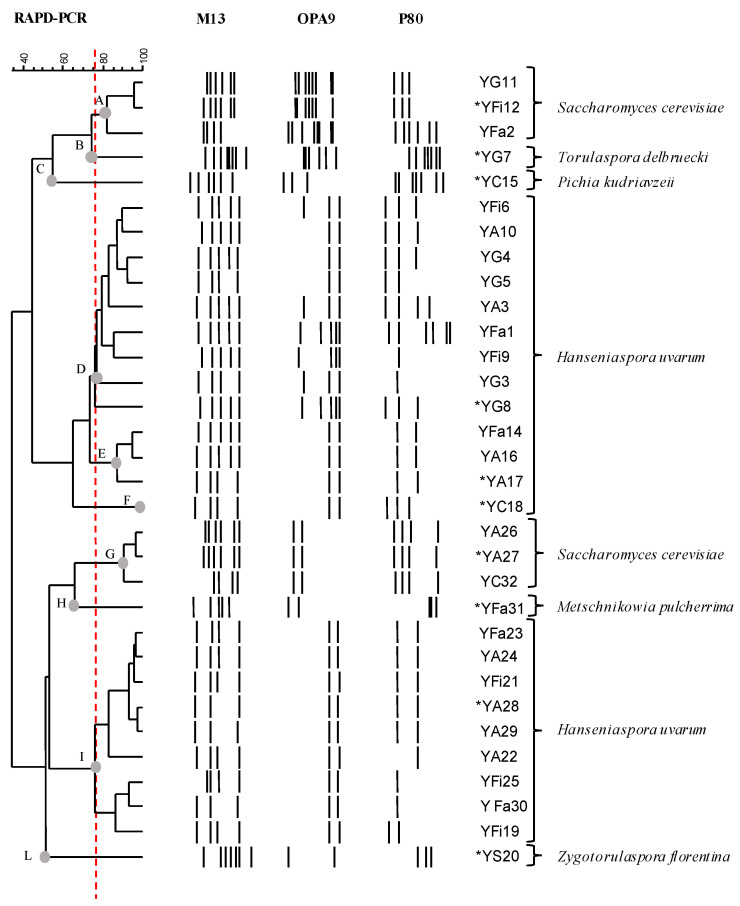
Dendrogram showing the similarity among RAPD-PCR profiles of 32 yeasts that were isolated from different wine-associated sources. The asterisks (*) indicate the strains that were identified by 26S rRNA sequencing analysis. The identification, based on blast comparison in GenBank, was reported in Appendix A.

Most yeasts were isolated from must and, to a lesser extent, from grapes of different varieties from the Campania Region. Almost all strains lacked decarboxylase activity, with the exception of four strains of *S. cerevisiae*, one of *Torulaspora delbrueckii*, and one of *Zygotorulaspora florentina*. The commercial strain *S. cerevisiae* LSC, used for comparison in dough fermentations, showed no potential decarboxylase activity against the amino acids that were tested. In this context, qualitative screening of decarboxylase activity becomes of technological interest to identify yeast strains that are potentially able to produce biogenic amines (Bas). In general, biogenic amines, mainly formed by microbes producing amino acid decarboxylase, are widely found in different foods such as meat, fish, cheese, and wine [26,27]. However, recent studies have also found detectable levels of BAs in sourdoughs where yeasts and lactic acid bacteria can contribute to the BAs formation [28]. Several authors have found that yeasts are also involved in the accumulation of BAs, but their role is debated and, for many aspects, controversial [23,27,29,30].

As highlighted by different authors [31,32], the ability of *S. cerevisiae* to produce biogenic amines seems to be strain-dependent and not a species-specific quality.

Therefore, the evaluation of the potential production of harmful compounds is important prior to their use for food fermentation. In this study, most of the NCY (with exception of one strain of *Torulaspora delbrueckii* and one of *Zygotorulaspora florentina*) lacked decarboxylase activity, suggesting their possible use in dough fermentation.

The strains were tested for leavening and acidification ability during 24 h of fermentation (Figure 2A,B). The highest leavening ability after 24 h, as expected, was observed in the commercial baker’s yeast (LSC; used as control) and in *S. cerevisiae* L12 and L27. Most of the NC-yeasts showed moderate leavening capacity after 6 h of incubation. After 24 h, however, the strains *H. uvarum* L8 and L13 showed a leaving capability that was comparable to that of S. cerevisiae L12, L27, and LCS. However, other *H. uvarum* strains (L23, L28, L16, L19, L25) were able to increase the dough volume to a greater extent than other *S. cerevisiae* strains (blue squares in Figure 2A).

It is well known that *Saccharomyces cerevisiae* rapidly converts sugars into ethanol and carbon dioxide [33] and due to its leavening ability, it is the most commonly used species in the bakery industry. However, in the last years, these attributes have also been found in some unconventional yeast species [11,34].

Zhou et al. (2017) [11] evaluated the effect of different NCYs on the volume increase and aroma profile of leavened doughs and model breads. They showed that the strain *Wickerhamomyces subpelliculosus* CBS5552 was able to produce and accumulate CO_2_ to a greater extent than *S. cerevisiae*; the strain *Kazachstania gamospora* CBS10400 also exhibited similar gas production to the control strain. *Torulaspora delbrueckii*, one of the most abundant NC-yeasts on the surface of grapes, has a great capability to ferment several carbon sources and its use as an alternative and efficient starter culture for leavened doughs has been demonstrated by several authors [10,15]; the co-culture of *S. cerevisiae* and *T. delbrueckii* was recently evaluated by Li et al. (2019) [34] and the results suggested that the mixture of strains increased CO_2_ production and volume of model doughs [34].

Interestingly, in our study, we found that some strains of *H. uvarum* are very robust and capable of fermenting dough in a similar manner to *S. cerevisiae* baker’s yeast. Hence, the need to characterize and select new NC-yeast strains with useful and interesting features for the bakery industry.

Figure 2B shows the results of the acidifying ability of yeast strains in the doughs after 6 and 24 h of fermentation. Most of the strains had moderate acidifying activity. After 24 h, the highest acidifying capability was observed in the control strain LSC and in *S. cerevisiae* L12, and in the strains *H. uvarum* L13 and L23.

From the analysis of leavening and acidifying capabilities, four strains of *H. uvarum* (L8, L13, L23, L28) and two strains of *S. cerevisiae* (L12 and L27) were selected for the subsequent leavening kinetics tests. Commercial yeast LSC was used for comparison purposes. The selected NCY showed poor leavening capacity after 6 h of incubation compared to the *S. cerevisiae* strains (Figure 3). After 24 h of rising, most of the *H. uvarum* strains (L13, L28, L8) resulted in a significant increase in the volume of the model doughs. Specifically, *H. uvarum* L13 reached a volume rise that was comparable to those of th control strain LSC and *S. cerevisiae* L12.

The strains *S. cerevisiae* L12, *H. uvarum* L8, L13, and L28 were selected on the basis of the best leavening kinetics and assessed for their ability to influence the volatile profile of dough after 24 h fermentation. Commercial baker’s yeast (LSC) was used as a control.

Table 2 reports 36 VOCs that were representative of six classes of compounds (aldehydes, ketones, esters and acetates, alcohols, acids, and terpenes) that characterized the different doughs. The results are reported as the relative peak area (RAP) for each dough.

Alcohol compounds were detected in a high amount in all samples. The main alcohol compounds that were detected were ethanol and isoamyl alcohol. Minor amounts of phenethyl alcohol, isobutanol, and 1-hexanol were also detected. The commercial culture (LSC) and *S. cerevisiae* L12 had the highest amount of ethanol, isobutanol, isoamyl alcohol, and phenethyl alcohol compared to the *H. uvarum* species. On the contrary, the *H. uvarum* strains (L8, L13, and L28) had the highest amount of 1-hexanol and 3-octanol compared to LSC and *S. cerevisiae* L12.

Also, a moderate production of esters and acetates was observed in all the samples. The dough that was inoculated with *S. cerevisiae* L12 recorded the highest amount of esters and acetates, mainly ethyl acetate and ethyl octanoate. The samples that were inoculated with *H. uvarum* strains were characterized for the presence of ethyl acetate, isoamyl acetate, and 4-methyl-2-pentyl acetate. Ethyl acetate was present in all the samples, although the amount of *H. uvarum* L13 was very low.

The metabolism of *S. cerevisiae* strains (LSC, L12) also produced acetoin, which, on the contrary, was almost completely absent in the NC-yeasts. The compound 2,3-butanedione was only found in the sample LSC while 2-octanone was found in all the samples.

Among terpenes, significant differences were found for the β-pinene and limonene levels, and *H. uvarum* L8, L13, and L28 resulted in the highest accumulation in the model doughs.

Aldehydes were recorded mainly in the samples that were inoculated with *S. cerevisiae* species (LSC and L12). The main aldehyde that was recorded was acetaldehyde, detected in significantly higher amounts in the LSC and L12 samples than in the doughs that were inoculated with *H. uvarum* species. Lower amounts of 2-methylpropanal and 3-methylbutanal were also detected in all the samples.

Acetic acid was detected in all samples but the highest amount was detected in the samples L12, LSC and L8.

To better understand the differences among the doughs that were inoculated with the different yeast strains, a PCA of the volatile compounds was performed (Figure 4). The two PCs explained the 71.68% of the total variance of the data. The doughs were located in three different zones of the plane. Regarding the score plot, a clear separation between doughs that were leavened with the *S. cerevisiae* species (negatively associated to PC1) and doughs that were inoculated with *H. uvarum* (positively associated to PC1) were evidenced. On the other hand, some differences were also found between the two *S. cerevisiae* strains, as the commercial starter was positively associated with PC2, while L12 had a negative correlation with PC2. The control sample (LSC) and *S. cerevisiae* L12 differed from *H. uvarum* strains mainly in alcohols (ethanol, isobutanol, isoamyl alcohol), ketones (acetoin), esters and acetates (ethyl butanoate, ethyl decanoate, ethyl hexanoate), and acid (acetic acid).

Zhou et al. (2017) [11] demonstrated that the model bread that was produced with NC-yeast belonging to *W. subpelliculosus* and *K. gamospora* species had a better aroma profile, characterized by buttery, nutty, and fruity tastes. Aslankoohi et al. (2016) [10] tested several NC-yeasts (that were isolated from different fermented foods and beverages) for the production of model leavened doughs and breads. The results demonstrated that samples that were inoculated with *Saccharomyces bayanus* and *Torulaspora delbrueckii* had an aromatic profile that was significantly different from that which was obtained in doughs and breads that were produced with commercial *S. cerevisiae* strain. Specifically, nutty and fruity tones, combined with satisfactory leavening efficiency, were recognized in breads that were produced with the two NC-species.

More recently, Liu et al. [11] demonstrated that the co-culture of *S. cerevisiae* and *T. delbrueckii* enhanced the organic acids and aminoacid production in leavened doughs.

The NC-yeasts that were tested in this study also imparted fruity, fresh, and herbal notes to the model doughs; additionally, the attributes that were related to alcohol fermentation (mainly ethanol) and organic acid production (vinegar note due to acetic acid content) are lower in NC-yeasts, suggesting that their use could enable the production of bakery products with sensory profile that is different from that which is obtained with standard commercial yeasts or with other *S. cerevisiae* strains.

## 4. Conclusions

This study provides an insight into the potential use of NC-yeasts in the bakery industry. Our data showed that selected strains of the *H. uvarum* species (L13, L28, L8) could be interesting candidates for the production of leavened doughs due to their ability to leaven and confer a different aroma profile to the doughs. In particular, these strains did not show any potential decarboxylase activity against the tested amino acids and reached after 24 h of fermentation a volume rise that was comparable to that of the *S. cerevisiae* control strain (LSC). Furthermore, volatile aroma of doughs that were fermented with the selected *H. uvarum* strains highly differed from those that were fermented with *S. cerevisiae* species mainly for alcohols, ketones, acids and esters, and acetates. So, the use of an alternative starter could increase the diversity of bakery products and meet the requirements of specific consumers.

However, further studies are needed to confirm the effectiveness of these strains in the production of baked goods.

## Figures and Tables

**Figure 2 microorganisms-10-01849-f002:**
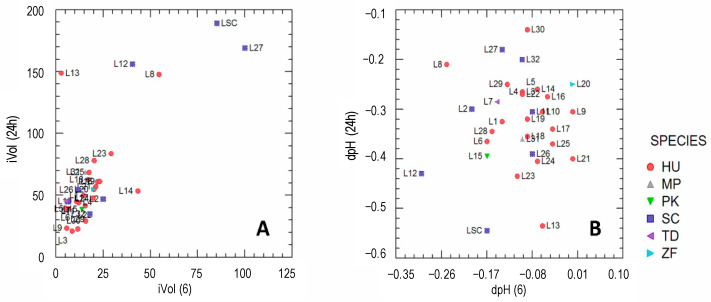
Leavening (volume increase of doughs, iVol; panel (**A**)) and acidifying (pH decrease of doughs, dpH; panel (**B**)) ability of yeast strains, after 24 h and 6 h of incubations. Symbols and colors indicate the membership to different species. HU, *Hanseniaspora uvarum*; MP, *Metschnikowia pulcherrima*; PK, *Pichia kudriavzevii*; SC, *Saccharomyces cerevisiae*; TD, *Torulaspora delbrueckii*; ZF, *Zygotorulaspora florentina.*

**Figure 3 microorganisms-10-01849-f003:**
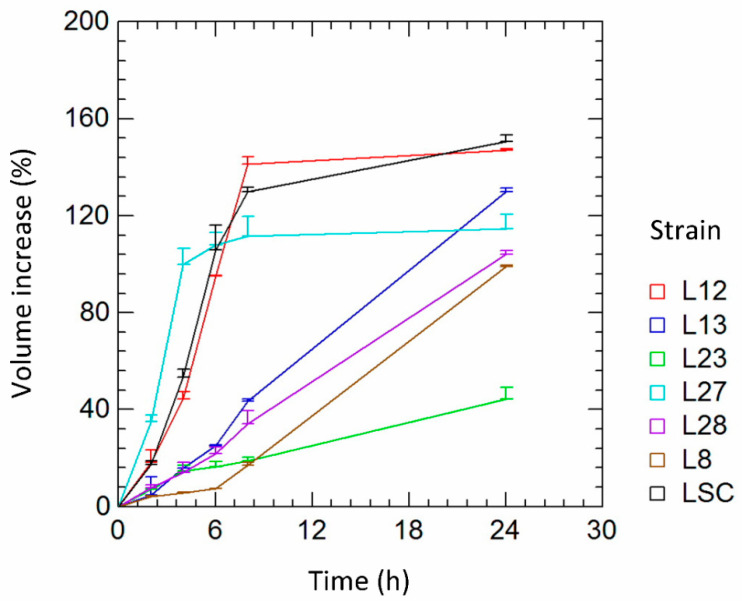
Leavening kinetics of the selected yeast strains. L12, L27: *S. cerevisiae*; L8, L13, L23, L27, and L28: *H. uvarum*; LSC: commercial yeast that was used as control.

**Figure 4 microorganisms-10-01849-f004:**
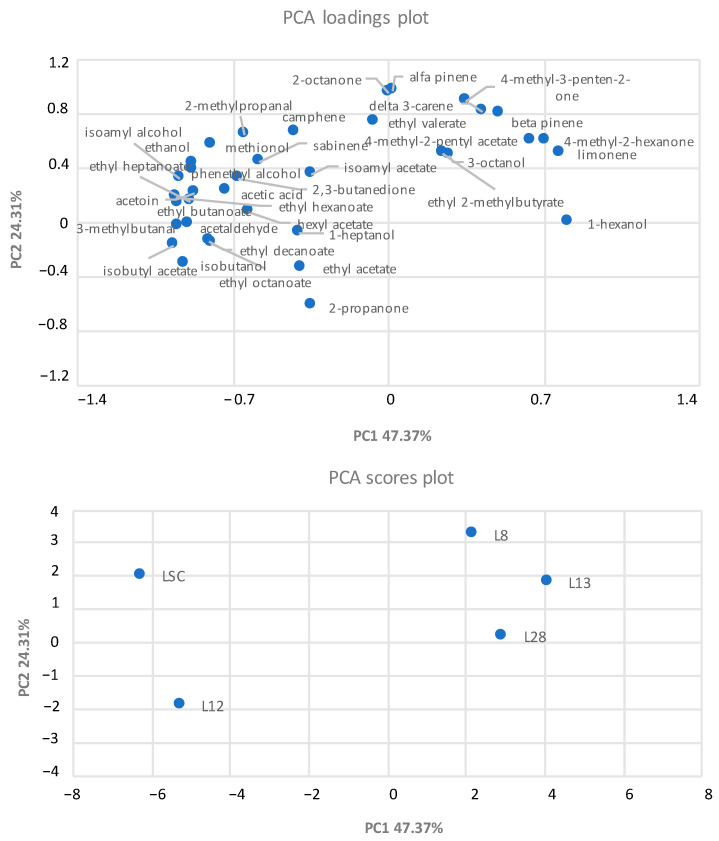
Principal component analysis (PCA) of volatile organic compounds that mainly differentiated the doughs inoculated with different yeasts strains. L12: *S. cerevisiae*; L8, L13, and L28: *H. uvarum*; LSC: commercial yeast used as control.

**Table 1 microorganisms-10-01849-t001:** The sources of isolation of the yeast strains and the results of the decarboxylase activity.

Strain	Sigle	Specie ^a^	Source ^b^	Variety ^c^	Decarboxylase Activity ^d^
YFi1	L1	*Hanseniaspora uvarum*	Grape	Fiano	No
YFa2	L2	*Saccharomyces cerevisiae*	Must	Falanghina	Arg, Leu
YA3	L3	*Hanseniaspora uvarum*	Must	Aglianico	No
YG4	L4	*Hanseniaspora uvarum*	Must	Greco	No
YG5	L5	*Hanseniaspora uvarum*	Must	Greco	No
YFi6	L6	*Hanseniaspora uvarum*	Must	Fiano	No
YG7	L7	*Torulaspora delbruekii*	Grape	Greco	Trp
YG8	L8	*Hanseniaspora uvarum*	Must	Greco	No
YFi9	L9	*Hanseniaspora uvarum*	Must	Fiano	No
YA10	L10	*Hanseniaspora uvarum*	Grape	Aglianico	No
YA11	L11	*Saccharomyces cerevisiae*	Must	Aglianico	No
YFi12	L12	*Saccharomyces cerevisiae*	Must	Fiano	No
YG13	L13	*Hanseniaspora uvarum*	Must	Greco	No
YFa14	L14	*Hanseniaspora uvarum*	Must	Falanghina	No
YC15	L15	*Pichia kudriavzevii*	Must	Coda di Volpe	Tyr
YA16	L16	*Hanseniaspora uvarum*	Must	Aglianico	No
YA17	L17	*Hanseniaspora uvarum*	Must	Aglianico	No
YC18	L18	*Hanseniaspora uvarum*	Grape	Coda di Volpe	No
YFi19	L19	*Hanseniaspora uvarum*	Must	Fiano passito	No
YS20	L20	*Zygotorulaspora florentina*	Grape	Sciascinoso	Arg, Leu, Phe, Tyr
YFi21	L21	*Hanseniaspora uvarum*	Grape	Fiano	No
YA22	L22	*Hanseniaspora uvarum*	Must	Aglianico	No
YFa23	L23	*Hanseniaspora uvarum*	Must	Falanghina	No
YA24	L24	*Hanseniaspora uvarum*	Must	Aglianico	No
YFi25	L25	*Hanseniaspora uvarum*	Must	Fiano	No
YA26	L26	*Saccharomyces cerevisiae*	Must	Aglianico	Arg, Leu, His, Phe, Tyr
YA27	L27	*Saccharomyces cerevisiae*	Must	Aglianico	His, Trp
YA28	L28	*Hanseniaspora uvarum*	Must	Aglianico	No
YA29	L29	*Hanseniaspora uvarum*	Must	Aglianico	No
YFa30	L30	*Hanseniaspora uvarum*	Grape	Falanghina	No
YFa31	L31	*Metschnikowia pulcherrima*	Must	Falanghina	No
YC32	L32	*Saccharomyces cerevisiae*	Must	Coda di Volpe	Leu, Tyr
Control	LSC	*Saccharomyces cerevisiae*	Commercial	-	No

^a^ Identification from Figure 1. ^b^ Source of isolation; ^c^ Vitis vinifera Cultivar; ^d^ Decarboxylase activity on YPD agar containing arginine (Arg), histidine (His), leucine (Leu), phenylalanine (Phe), tryptophan (Trp), tyrosine (Tyr).

**Table 2 microorganisms-10-01849-t002:** Profiles of volatile organic compounds that were detected in doughs inoculated with different yeast strains.

RI	Compounds	Yeast Strains					Odor ^#^
		LSC	L12	L8	L13	L28	
	**Aldehydes**						
719	acetaldehyde	5.37 ± 0.02 ^a^	5.23 ± 0.43 ^a^	3.02 ± 0.05 ^b^	3.35 ± 0.22 ^b^	3.78 ± 0.36 ^b^	fruity
853	2-methylpropanal	0.23 ± 0.00 ^a^	0.12 ± 0.01 ^b^	0.12 ± 0.00 ^bc^	0.11 ± 0.00 ^bd^	0.13 ± 0.00 ^b^	banana, sweet
976	3-methylbutanal	0.25 ± 0.01 ^a^	0.40 ± 0.02 ^b^	0.13 ± 0.00 ^c^	0.04 ± 0.00 ^d^	0.04 ± 0.00 ^d^	aldehydic
	*tot*	5.84 ± 0.04 ^a^	5.75 ± 0.47 ^a^	3.26 ± 0.06 ^b^	3.50 ± 0.22 ^b^	3.95 ± 0.36 ^b^	
	**Ketones**						
890	2-propanone	nd	0.30 ± 0.02 ^a^	nd	nd	0.12 ± 0.01 ^b^	ethereal, apple
963	2,3-butanedione	1.04 ± 0.01 ^a^	nd	nd	nd	nd	butter, fatty
1124	4-methyl-2-hexanone	0.20 ± 0.01 ^a^	0.23 ± 0.02 ^b^	0.44 ± 0.04 ^c^	0.57 ± 0.06 ^d^	0.45 ± 0.01 ^c^	fruity
1140	4-methyl-3-penten-2-one	0.35 ± 0.02 ^a^	0.30 ± 0.02 ^b^	0.49 ± 0.05 ^c^	0.40 ± 0.02 ^d^	0.42 ± 0.03 ^c^	vegetable
1310	acetoin	14.52 ± 0.38 ^a^	5.15 ± 0.32 ^b^	0.74 ± 0.03 ^c^	0.31 ± 0.03 ^d^	0.48 ± 0.04 ^e^	sweet, butter
1490	2-octanone	2.14 ± 0.21 ^a^	1.92 ± 0.18 ^a^	2.29 ± 0.22 ^a^	2.37 ± 0.15 ^a^	2.00 ± 0.12 ^a^	earthy, grass
	*tot*	18.25 ± 0.16 ^a^	7.60 ± 0.43 ^b^	3.95 ± 0.18 ^c^	3.66 ± 0.22 ^c^	3.36 ± 0.18 ^c^	
	**Esters and acetates**						
1134	ethyl valerate	0.40 ± 0.04 ^a^	0.07 ± 0.00 ^b^	0.40 ± 0.03 ^a^	0.13 ± 0.01 ^c^	0.36 ± 0.03 ^a^	sweet, fruity
905	ethyl acetate	16.02 ± 1.08 ^a^	25.79 ± 1.60 ^b^	18.92 ± 1.40 ^c^	6.11 ± 0.46 ^d^	22.65 ± 2.01 ^b^	fruity, sweet
1000	isobutyl acetate	0.54 ± 0.03 ^a^	0.47 ± 0.02 ^b^	0.20 ± 0.02 ^c^	0.14 ± 0.01 ^d^	0.24 ± 0.01 ^e^	sweet, fruity
1015	ethyl butanoate	0.55 ± 0.03 ^a^	0.39 ± 0.02 ^b^	0.15 ± 0.01 ^c^	0.18 ± 0.02 ^c^	0.16 ± 0.01 ^c^	sweet, fruity
1057	ethyl 2-methylbutyrate	nd	nd	0.42 ± 0.02 ^a^	nd	nd	sweet, fruity
1118	4-methyl-2-pentyl acetate	0.67 ± 0.01 ^a^	0.76 ± 0.01 ^b^	0.92 ± 0.04 ^c^	1.07 ± 0.11 ^c^	0.90 ± 0.02 ^c^	sweet, fruity
1130	isoamyl acetate	1.60 ± 0.16 ^a^	2.93 ± 0.04 ^b^	2.56 ± 0.25 ^c^	1.64 ± 0.11 ^a^	1.76 ± 0.14 ^a^	sweet, fruity
1260	ethyl hexanoate	2.08 ± 0.02 ^a^	2.10 ± 0.17 ^a^	0.70 ± 0.05 ^b^	0.55 ± 0.02 ^c^	0.48 ± 0.00 ^d^	sweet, fruity
1298	hexyl acetate	0.12 ± 0.01 ^a^	0.14 ± 0.1 ^a^	0.14 ± 0.01 ^a^	0.07 ± 0.01 ^b^	0.07 ± 0.00 ^b^	sweet, fruity
1337	ethyl heptanoate	0.20 ± 0.00 ^a^	0.26 ± 0.0 ^b^	0.11 ± 0.01 ^c^	0.10 ± 0.00 ^c^	0.06 ± 0.01 ^d^	fruity
1500	ethyl octanoate	1.45 ± 0.10 ^a^	4.15 ± 0.35 ^b^	0.22 ± 0.01 ^c^	0.15 ± 0.01 ^d^	0.11 ± 0.00 ^e^	fruity, wine
1611	ethyl decanoate	0.35 ± 0.03 ^a^	0.92 ± 0.05 ^b^	0.07 ± 0.00 ^c^	0.07 ± 0.00 ^c^	0.03 ± 0.00 ^d^	sweet, waxy
	*tot*	23.98 ± 0.93 ^a^	37.97 ± 2.15 ^b^	24.81 ± 1.52 ^a^	10.19 ± 0.69 ^c^	26.82 ± 1.90 ^d^	
	**Alcohols**						
989	ethanol	122.75 ± 0.60 ^a^	113.62 ± 1.83 ^b^	86.39 ± 1.23 ^c^	84.61 ± 4.75 ^c^	79.00 ± 6.86 ^c^	alcohol
1120	isobutanol	7.65 ± 0.21 ^a^	8.00 ± 0.62 ^a^	5.58 ± 0.46 ^b^	4.57 ± 0.41 ^c^	5.75 ± 0.30 ^b^	ethereal
1230	isoamyl alcohol	61.60 ± 0.02 ^a^	58.59 ± 5.08 ^a^	47.77 ± 1.07 ^c^	42.08 ± 4.01 ^d^	43.87 ± 2.58 ^d^	fruity
1395	1-hexanol	6.47 ± 0.33 ^a^	5.75 ± 0.51 ^a^	8.49 ± 0.92 ^c^	7.20 ± 0.65 ^c^	7.96 ± 0.58 ^c^	green, fruity
1410	3-octanol	1.02 ± 0.01 ^a^	1.94 ± 0.10 ^b^	2.50 ± 0.19 ^c^	2.68 ± 0.05 ^c^	0.93 ± 0.04 ^d^	earthy, mushroom
1453	1-heptanol	0.91 ± 0.05 ^a^	nd	nd	nd	0.44 ± 0.02 ^b^	sweet, woody
1695	methionol	0.20 ± 0.01 ^a^	0.13 ± 0.01 ^b^	0.13 ± 0.00 ^b^	0.08 ± 0.00 ^c^	0.08 ± 0.00 ^c^	sulfurous, onion
1925	phenethyl alcohol	10.19 ± 0.16 ^a^	10.00 ± 1.06 ^a^	5.74 ± 0.29 ^b^	4.27 ± 0.03 ^c^	2.30 ± 0.10 ^d^	floral, rose
	*tot*	210.77 ± 1.0 ^a^	198.04 ± 1.15 ^b^	156.60 ± 3.78 ^c^	145.48 ± 0.25 ^d^	140.34 ± 4.72 ^d^	
	**Acids**						
1510	acetic acid	2.52 ± 0.03 ^a^	4.67 ± 0.48 ^b^	2.84 ± 0.09 ^c^	0.84 ± 0.08 ^d^	0.43 ± 0.02 ^e^	sharp, vinegar
	**Terpenes**						
1008	alfa pinene	0.84 ± 0.03 ^a^	0.39 ± 0.03 ^b^	0.88 ± 0.07 ^a^	0.79 ± 0.05 ^a^	0.56 ± 0.04 ^c^	piney, woody
1050	camphene	0.31 ± 0.01 ^a^	0.16 ± 0.01 ^b^	0.23 ± 0.00 ^c^	0.19 ± 0.02 ^d^	0.20 ± 0.02 ^d^	woody, herbal
1112	beta pinene	14.94 ± 0.63 ^a^	14.31 ± 1.12 ^a^	23.27 ± 0.03 ^b^	23.16 ± 0.57 ^b^	23.17 ± 2.36 ^b^	fresh, green
1126	sabinene	0.41 ± 0.00 ^a^	0.08 ± 0.00 ^b^	0.16 ± 0.01 ^c^	0.07 ± 0.01 ^b^	0.15 ± 0.01 ^c^	woody, citrus
1133	delta 3-carene	0.50 ± 0.02 ^a^	0.50 ± 0.04 ^a^	0.81 ± 0.03 ^b^	0.83 ± 0.01 ^b^	0.80 ± 0.07 ^b^	citrus, herbal
1223	limonene	0.53 ± 0.02 ^a^	0.83 ± 0.06 ^b^	1.70 ± 0.12 ^c^	1.81 ± 0.00 ^d^	1.82 ± 0.16 ^cd^	citrus
	*tot*	17.54 ± 0.59 ^a^	16.27 ± 1.23 ^a^	27.04 ± 0.14 ^b^	26.84 ± 0.53 ^b^	26.69 ± 2.66 ^b^	

RI = retention index. RIs were calculated with van Den Dool and Kratz formula. Calculated RIs were compared using the online NIST database (http://webbook.nist.gov/chemistry/; accessed on 1 October 2021) for high polar column for InnoWAX or similar stationary phases. All the compounds were identified by the matching RI and MS. The results are expressed as RAP = relative peak area (area peak compound/area peak internal standard) × 100. Lowercase letters (a, b, c, d, e) indicate significant differences (*p* < 0.05) relative volatile compounds among the five different doughs. Each value is expressed as mean ± SD of two replicates. nd = not detected. ^#^ based on online databases (www.flavornet.org; accessed on 1 October 2021, and www.thegoodscentscompany.com; accessed on 1 October 2021).

## Data Availability

The data that are presented in this study are available on request from the corresponding author.

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
