# Peer review of "Selection of Non-Saccharomyces Wine Yeasts for the Production of Leavened Doughs"

_microorganisms, 2022, doi:10.3390/microorganisms10091849_

Round 1
Reviewer 1 Report
The manuscript investigated if non-conventional yeast (NCY) strains can be used as an alternative in leavened doughs and if they produce interesting aromatic compounds. The authors used a combination of RAPD-PCR profiling and 26S rRNA region to identify non-conventional yeast strains. The authors screened for decarboxylase activity to determine what strains produce biogenic amines. The authors then screened all strains for leavening capability by measuring dough area after growth periods. Follow this, the authors took the best performing NCY and repeated the leavening capability measurements. Finally, the authors used SPME-GC/MS to measure volatile organic compounds.
This manuscript is well written and clearly describes the field of research, the experimental questions, and the methods used. The results and discussion are detailed, well written, and clearly presented. Some additional and adjustments to figures would help engage the reader. Overall, an interesting and well performed study.
Major Comments:
1. For volatile organic compounds (VOC) analysis, why was only two replicates used? For true statistical comparisons between strains at least three replicates would be required.
Minor Comments:
1. While it is completely reasonable to use RAPD followed by 26S rRNA sequencing on select strains. Why did the authors not just perform 26S rRNA or ITS sequencing on all strains? This would remove any ambiguity if strains clustered by RAPD but in fact were different species.
2. Table 1: In addition to this table, photos of decarboxylase activity on agar would visually help the reader see the difference in decarboxylase activity.
3. Figure 2: While the scatter plots are completely appropriate, it may help the reader to show dVol on a bar graph as well.
4. Table 2: It would be interesting and benefit the reader if any compounds of interest were also plotted as bar graphs to highlight how different VOCs are between strains.

Author Response
We are very grateful to the Reviewer for the constructive comments which have effectively contributed to improve our paper.
The manuscript was modified according to the reviewers’ suggestions. A detailed point-by-point response to the Reviewers’ suggestions is provided.
We hope that the new version of the manuscript will now be suitable for publication in “Microorganisms” Journal. Below our detailed responses. The changes made to the manuscript are highlighted by using a different color (yellow).
REVIEWER 1
Reviewer 1: The manuscript investigated if non-conventional yeast (NCY) strains can be used as an alternative in leavened doughs and if they produce interesting aromatic compounds. The authors used a combination of RAPD-PCR profiling and 26S rRNA region to identify non-conventional yeast strains. The authors screened for decarboxylase activity to determine what strains produce biogenic amines. The authors then screened all strains for leavening capability by measuring dough area after growth periods. Follow this, the authors took the best performing NCY and repeated the leavening capability measurements. Finally, the authors used SPME-GC/MS to measure volatile organic compounds.
This manuscript is well written and clearly describes the field of research, the experimental questions, and the methods used. The results and discussion are detailed, well written, and clearly presented. Some additional and adjustments to figures would help engage the reader. Overall, an interesting and well performed study.
Answer: Thank you very much for your comment.
Reviewer 1: For volatile organic compounds (VOC) analysis, why was only two replicates used? For true statistical comparisons between strains at least three replicates would be required.
Answer: We are aware that a higher number of replicates may provide greater statistical significance. However, we believe that in this study, which is a preliminary investigation, two replicates may be useful to provide indicative information that can be exploit as pilot experiments to drive further trials with a more complex experimental design and a high number of replication. Furthermore, we are competent with SPME-GC/MS analysis on dough matrices (as demonstrated by several papers), so we are confident of the faithfulness of our results.
Reviewer 1: While it is completely reasonable to use RAPD followed by 26S rRNA sequencing on select strains. Why did the authors not just perform 26S rRNA or ITS sequencing on all strains? This would remove any ambiguity if strains clustered by RAPD but in fact were different species.
Answer: The referee is right in pointing that sequencing is the best way to correctly identify a strain. However, RAPD-PCR assays offer a robust and inexpensive approach for routine observations and can be considered a good screening method. In our work, the choice of strains to be sequenced was made within clusters with a high similarity coefficient (75%), allowing strains reasonably belonging to the same species to be grouped together. On the other hand, also the previous phenotypic tests carried out on our strains gave an indication of the membership to the different species, successively confirmed with molecular methods in this study.
Reviewer 1: Table 1: In addition to this table, photos of decarboxylase activity on agar would visually help the reader see the difference in decarboxylase activity.
Answer: We do not have pictures of agar plates, so we cannot supply. However, the description of result interpretation (“The YPD plates were incubated for 7 days at 28 °C, and the appearance of purple halos (positive results) around the colonies was checked at 24h intervals”) was already reported in Material and Methods sections.
Reviewer 1: Figure 2: While the scatter plots are completely appropriate, it may help the reader to show dVol on a bar graph as well.
Answer: We believe that the scatterplot gives a better indication of the distribution of yeast strains, based on the tested properties (dVol and dpH after 6h and 24h of incubation), compared to the bar plots. Furthermore, the high number of strains would generate very thin bars, which could compromise the correct visualization by the readers. In fact, while the proposed scatterplots allow to enclose 4 variables (dVol, 6h; dVol, 24 h; dpH, 6h; dpH, 24 h) in only 2 graphs, the use of bar plots would require 4 separate graphs. The merging of bar graphs, in fact, would compromise even more the visualization of results (very thin bars).
Reviewer 1: Table 2: It would be interesting and benefit the reader if any compounds of interest were also plotted as bar graphs to highlight how different VOCs are between strains.
Answer: We believe that adding another graph on VOC compounds (a table and a PCA plot were already provided) would be redundant.
Reviewer 2 Report
The manuscript "Selection of non-Saccharomyces wine yeasts for the production of leavened doughs" presents interesting results and preliminary results regarding the use of non-conventional yeasts from the wine industry as starters in bakery products.
The article is well written, but some improvements can still be implemented. In the introduction, the authors could mention that not only yeasts are used in bakery products but also lactic acid bacteria (i.e. 10.3389/fbioe.2022.888827, 10.1080/15428052.2019.1680472 ).
In the results section, please correct the microorganism names to italics. Revise the whole article.
The article should also be revised for some grammatical errors i.e. Alcohols compounds were detected in the high amount in all samples. > Alcohol compounds were detected in a high amount in all samples.
line 310 - negatively associate to > negatively associated to
line 340 - on the bakery industry > in the bakery industry
line 343 - The leavening agents have great impact > The leavening agents greatly impact
Revise the whole article for clarity.
Otherwise, the material and method section is well written and easy to reproduce, and the results are well discussed. After some corrections, the manuscript can be considered for publication.
Author Response
We are very grateful to the Reviewers for the constructive comments which have effectively contributed to improve our paper.
The manuscript was modified according to the reviewers’ suggestions. A detailed point-by-point response to the Reviewers’ suggestions is provided.
We hope that the new version of the manuscript will now be suitable for publication in “Microorganisms” Journal. Below our detailed responses. The changes made to the manuscript are highlighted by using a different color (yellow).
REVIEWER 2
The manuscript "Selection of non-Saccharomyces wine yeasts for the production of leavened doughs" presents interesting results and preliminary results regarding the use of non-conventional yeasts from the wine industry as starters in bakery products.
Reviewer 2: The article is well written, but some improvements can still be implemented. In the introduction, the authors could mention that not only yeasts are used in bakery products but also lactic acid bacteria (i.e. 10.3389/fbioe.2022.888827, 10.1080/15428052.2019.1680472).
Answer: Thank you for your suggestion. We included the reference 10.1080/15428052.2019.1680472 that better fits the topic of the manuscript.
Reviewer 2: In the results section, please correct the microorganism names to italics. Revise the whole article.
Answer: Thank you, we checked the microorganism names in the entire manuscript as suggested.
Reviewer 2: The article should also be revised for some grammatical errors i.e. Alcohols compounds were detected in the high amount in all samples. > Alcohol compounds were detected in a high amount in all samples.
Answer: Done. Other grammatical or typing errors have been revised.
Reviewer 2: line 310 - negatively associate to > negatively associated to
Answer: Done.
Reviewer 2: line 340 - on the bakery industry > in the bakery industry
Answer: Done.
Reviewer 2: line 343 - The leavening agents have great impact > The leavening agents greatly impact
Answer: Done.
Reviewer 2: Revise the whole article for clarity.
Answer: Thank you for the suggestion. The manuscript has been carefully reviewed.
Reviewer 2: Otherwise, the material and method section is well written and easy to reproduce, and the results are well discussed. After some corrections, the manuscript can be considered for publication.
Answer: Thank you for your comment. All corrections were made as suggested.
Reviewer 3 Report
The paper proposed by Zotta and her team is very interesting revealing a very important topic regarding using of non-Saccharomyces yeasts as alternative starters to promote biodiversity and quality of fermented foods and beverages.
They identified a number of wine-associated yeasts and analyzed for decarboxylase activity and leavening ability, comparatively with a commercial Saccharomyces cerevisiae strain. Species-level identification was carried out using BLAST comparison on GenBank database. Most of the strains identified belonged to the species Hanseniaspora uvarum, while some isolates matched to the species S. cerevisiae and there some isolated strains belonging to other species.
The most promising strains were analyzed for the decarboxylase activity of amino acids for the leavening capability in model wheat doughs. Also, the researchers evaluated the volatile organic compounds in model wheat doughs, produced by the selected strains. The chemical analysis of the compounds resulted after the fermentation processes using different species of identified yeasts, reveals some similarities with the Saccharomyces cerevisiae control strain.
The study shows that some H. uvarum strains could be used for the leavened doughs, because their abilities to produces a different aroma profile to the doughs, modifying the sensory perception of the bakery products.
The work is well presented and structured. The methodology, results and discussion section are very well documented.
The paper could be read by a native speaker to ensure the correct use of English in few points, however there are no major grammar mistakes that would make the text difficult to comprehend. The authors should write the name of the species with italic font (e.g. lines 193,194,195,196, 205, 214, 220 and so on, in all the document. Lines 246-249 -the authors should reformulate the phrase.
Author Response
We are very grateful to the Reviewer for the constructive comments which have effectively contributed to improve our paper.
The manuscript was modified according to the reviewers’ suggestions. A detailed point-by-point response to the Reviewers’ suggestions is provided.
We hope that the new version of the manuscript will now be suitable for publication in “Microorganisms” Journal. Below our detailed responses. The changes made to the manuscript are highlighted by using a different color (yellow).
REVIEWER 3
Reviewer 3: The paper proposed by Zotta and her team is very interesting revealing a very important topic regarding using of non-Saccharomyces yeasts as alternative starters to promote biodiversity and quality of fermented foods and beverages.
They identified a number of wine-associated yeasts and analyzed for decarboxylase activity and leavening ability, comparatively with a commercial Saccharomyces cerevisiae strain. Species-level identification was carried out using BLAST comparison on GenBank database. Most of the strains identified belonged to the species Hanseniaspora uvarum, while some isolates matched to the species S. cerevisiae and there some isolated strains belonging to other species.
The most promising strains were analyzed for the decarboxylase activity of amino acids for the leavening capability in model wheat doughs. Also, the researchers evaluated the volatile organic compounds in model wheat doughs, produced by the selected strains. The chemical analysis of the compounds resulted after the fermentation processes using different species of identified yeasts, reveals some similarities with the Saccharomyces cerevisiae control strain.
The study shows that some H. uvarum strains could be used for the leavened doughs, because their abilities to produces a different aroma profile to the doughs, modifying the sensory perception of the bakery products.
The work is well presented and structured. The methodology, results and discussion section are very well documented.
Answer: Thank you very much for your comment.
Reviewer 3: The paper could be read by a native speaker to ensure the correct use of English in few points, however there are no major grammar mistakes that would make the text difficult to comprehend. The authors should write the name of the species with italic font (e.g. lines 193,194,195,196, 205, 214, 220 and so on, in all the document. Lines 246-249 -the authors should reformulate the phrase.
Answer: We thank the reviewer for his suggestion. The English language was accurately reviewed. All the suggestions were modified accordingly.
Reviewer 4 Report
The manuscript entitled “Selection of non-Saccharomyces wine yeasts for the production of leavened doughs” deals with the potential use of non-conventional yeasts in the bakery industry. They have identified a yeast species (among 32 isolated strains) that could be interesting for the production of leavened doughs, conferring different aromas. The manuscript is well written (despite English mistakes) and very clear. It is of great importance for researchers in the area, as well as for the food industry.
Some points should be addressed and/or corrected:
Introduction section:
- Lines 66-70: the term “on the other hand” is confusing here. Don’t you mean “moreover”?
- Also, the authors depict the importance of NCY to bakery products, reporting some studies with those yeasts. However, they should also depict the innovation of the present study. What will we see in the present study that hasn’t been performed yet? This should be clear in the introduction section before the paragraph related to the goal of the study.
Methodology
- Acronyms should be defined When mentioned for the first time, for example, RAPD-PCR
- Section 2.1) The isolation procedure should be briefly described.
- table 1 should be closer to the paragraph mentioning it for the first time
-Section 2.3) A control (without the addition of the amino acid) was performed? Since peptone presents amino acids…
Results
Several species names are not in italics, and they should be
“In Table 1 are also reported the results of the decarboxylase activity of the yeasts.” Consider rewriting the sentence.
“Several authors in fact found that also yeasts are involved...” consider changing to: Several authors, in fact, found that yeasts are also involved...
Table 1: The result of the control yeast is missing.
Figure 2: Legend should help readers understand the graphics. What dpH and dVol mean? The numbers represent...; 24 h, 6h?
Lines 239-241: references should be cited here.
“Hence, the need to investigate the potential of different NC-yeasts for use in breadmaking, since only a reduced number of these species have been described.” Consider rewriting this sentence
Lines 262-266: This is related to results in Figure 3? If yes, please refer to Figure 3
Lines 274-277: consider reviewing the text.
Conclusions
Last paragraph, lines 346-348: this sentence is not a conclusion of your work.
Author Response
We are very grateful to the Reviewer for the constructive comments which have effectively contributed to improve our paper.
The manuscript was modified according to the reviewers’ suggestions. A detailed point-by-point response to the Reviewers’ suggestions is provided.
We hope that the new version of the manuscript will now be suitable for publication in “Microorganisms” Journal. Below our detailed responses. The changes made to the manuscript are highlighted by using a different color (yellow).
REVIEWER 4
The manuscript entitled “Selection of non-Saccharomyces wine yeasts for the production of leavened doughs” deals with the potential use of non-conventional yeasts in the bakery industry. They have identified a yeast species (among 32 isolated strains) that could be interesting for the production of leavened doughs, conferring different aromas. The manuscript is well written (despite English mistakes) and very clear. It is of great importance for researchers in the area, as well as for the food industry. Some points should be addressed and/or corrected:
Answer: We thank the reviewer for his kind comment. All corrections were done, accordingly.
Introduction section:
Reviewer 4: Lines 66-70: the term “on the other hand” is confusing here. Don’t you mean “moreover”?
Answer: The reviewer is right. The correction was done.
Reviewer 4: Also, the authors depict the importance of NCY to bakery products, reporting some studies with those yeasts. However, they should also depict the innovation of the present study. What will we see in the present study that hasn’t been performed yet? This should be clear in the introduction section before the paragraph related to the goal of the study.
Answer: Thank for the suggestion. This aspect was clarified in the introduction section.
Methodology
Reviewer 4: Acronyms should be defined When mentioned for the first time, for example, RAPD-PCR
Answer: Done.
Reviewer 4: Section 2.1) The isolation procedure should be briefly described.
Answer: We thank the referee for his comment. However, the strains used in the study were already present in the CNR microbial collection (as already reported in section 2.1). Therefore, I did not consider it useful to add the isolation methods. I revised this part by specifying in the text that the strains had been “previously” isolated.
Reviewer 4: table 1 should be closer to the paragraph mentioning it for the first time
Answer: As suggested the table was moved above.
Reviewer 4: Section 2.3) A control (without the addition of the amino acid) was performed? Since peptone presents amino acids…
Answer: In the optimization step, the unsupplemented YPD (without amino acids) was used to test the efficacy in yeast growth and the adequacy in the color change of the pH-indicator bromocresol purple (BCP), after different days of incubation. Therefore, although the YPD medium contains a pool of aminoacid-based components, we are confident that the color change of BCP was exclusively due to the decarboxylase activity of the strains. However, to improve the understanding by readers, we have added a short sentence in section 2.3.
Results
Reviewer 4: Several species names are not in italics, and they should be
Answer: Thank you, we checked the microorganism names in the entire manuscript as suggested.
Reviewer 4: “In Table 1 are also reported the results of the decarboxylase activity of the yeasts.” Consider rewriting the sentence.
Answer: The sentence was rewritten.
Reviewer 4: “Several authors in fact found that also yeasts are involved...” consider changing to: Several authors, in fact, found that yeasts are also involved...
Answer: Done.
Reviewer 4: Table 1: The result of the control yeast is missing.
Answer: The commercial yeast LSC (control) was also tested for decarboxylase activity (as already indicated in section 2.3), but it was not reported in Table 1 as the latter includes only the data relating to our strains. However, a revised version of the Table 1, which also includes the decarboxylase activity of LSC, has been provided. Results of enzymatic activity of LSC have been also reported in “Results and discussion section”.
Reviewer 4: Figure 2: Legend should help readers understand the graphics. What dpH and dVol mean? The numbers represent...; 24 h, 6h?
Answer: The legend of Figure 2 has been revised to improve the meaning of results and the understanding by readers. Additionally, labels of X- and Y-axes of panel A have been changed as iVol (volume increase). Figure 2 has been replaced.
Reviewer 4: Lines 239-241: references should be cited here.
Answer: The references were cited.
Reviewer 4: “Hence, the need to investigate the potential of different NC-yeasts for use in breadmaking, since only a reduced number of these species have been described.” Consider rewriting this sentence
Answer: The sentence was rewritten.
Reviewer 4: Lines 262-266: This is related to results in Figure 3? If yes, please refer to Figure 3
Answer: The sentence was referred to Fig.3.
Reviewer 4: Lines 274-277: consider reviewing the text.
Answer: The sentence was rewritten.
Conclusions
Reviewer 4: Last paragraph, lines 346-348: this sentence is not a conclusion of your work.
Answer: Thank you for the right suggestion. The sentence was moved to the introduction section.
Round 2
Reviewer 2 Report
The authors implemented some of the specified corrections, but some of them were totally ignored. With so many errors the manuscript can't be accepted for publication, only after thorough revisions:
- please leave a space between the number and degree sign (i.e. lines 101, 103, 105, 178...) revise the whole manuscript
- the name of the microorganisms were not corrected to italics, maybe some of them, but the authors need to revise them once again in the manuscript ( i.e. lines 202-205, 217, 218, 232, and so on)
- please leave a space between the number and h - (i.e. line 235, 268)
- in figure 3 the x-axes should end at 24 h
- the conclusion section should be described in more detail.
Also, based on the common scientific and grammatical errors, the manuscript must be thoroughly revised by all the authors and a native English speaker to be accepted for publication.
Author Response
RESPONSE TO REVIEWERS
Dear Editor, we are very grateful to the Reviewer for the suggestions.
The manuscript was modified accordingly. A detailed point-by-point response to the Reviewers’ suggestions is provided.
With the first revision, as can be seen from the .doc format, the names of the micro-organisms had all been corrected in italics as requested, but we do not understand why they were lost with the conversion to pdf.
So, again we have changed them. We apologize for the oversight.
We hope that the new version of the manuscript will now be suitable for publication in “Microorganisms” Journal. Below our detailed responses. The changes made to the manuscript are highlighted by using a different color (yellow).
REVIEWER 2
Reviewer 2: The authors implemented some of the specified corrections, but some of them were totally ignored. With so many errors the manuscript can't be accepted for publication, only after thorough revisions:
Answer: are very grateful to the Reviewer for the suggestions.
The manuscript was modified accordingly. A detailed point-by-point response is provided.
Reviewer 2: - please leave a space between the number and degree sign (i.e. lines 101, 103, 105, 178...) revise the whole manuscript
Answer: Done.
Reviewer 2: - the name of the microorganisms were not corrected to italics, maybe some of them, but the authors need to revise them once again in the manuscript ( i.e. lines 202-205, 217, 218, 232, and so on)
Answer: With the first revision, as can be seen from the .doc format, the names of the micro-organisms had all been corrected in italics as requested, but we do not understand why they were lost with the conversion to pdf. So, again we have changed them. We apologize for the oversight.
Reviewer 2: - please leave a space between the number and h - (i.e. line 235, 268)
Answer: Done.
Reviewer 2: - in figure 3 the x-axes should end at 24 h
Answer: We decided to set the end of the x-axis at 30 hours to improve the graph visualization and the understanding by the readers. In fact, if we set the x-axis to 24 h, the last points and the standard deviation bars are not correctly visible because they overlap the y-axis (please find figure below).
Therefore, we believe that the first version of the graph is more appropriate.
Reviewer 2: - the conclusion section should be described in more detail.
Answer: The conclusion was ameliorated, accordingly.
Reviewer 2: Also, based on the common scientific and grammatical errors, the manuscript must be thoroughly revised by all the authors and a native English speaker to be accepted for publication.
Answer: We thank the reviewer for his suggestion. The manuscript was carefully reviewed for the English language.
See the attached file
